# Bottom-Up Cu Filling of High-Aspect-Ratio through-Diamond vias for 3D Integration in Thermal Management

**DOI:** 10.3390/mi14020290

**Published:** 2023-01-22

**Authors:** Kechen Zhao, Jiwen Zhao, Xiaoyun Wei, Xiaoyu Guan, Chaojun Deng, Bing Dai, Jiaqi Zhu

**Affiliations:** 1National Key Laboratory of Science and Technology on Advanced Composites in Special Environments, Harbin Institute of Technology, Harbin 150080, China; 2Huawei Technologies Co., Ltd., Dongguan 523799, China

**Keywords:** through-diamond vias, prewetting, electroplating, copper filling, 3D integration, thermal management

## Abstract

Three-dimensional integrated packaging with through-silicon vias (TSV) can meet the requirements of high-speed computation, high-density storage, low power consumption, and compactness. However, higher power density increases heat dissipation problems, such as severe internal heat storage and prominent local hot spots. Among bulk materials, diamond has the highest thermal conductivity (≥2000 W/mK), thereby prompting its application in high-power semiconductor devices for heat dissipation. In this paper, we report an innovative bottom-up Cu electroplating technique with a high-aspect-ratio (10:1) through-diamond vias (TDV). The TDV structure was fabricated by laser processing. The electrolyte wettability of the diamond and metallization surface was improved by Ar/O plasma treatment. Finally, a Cu-filled high-aspect-ratio TDV was realized based on the bottom-up Cu electroplating process at a current density of 0.3 ASD. The average single-via resistance was ≤50 mΩ, which demonstrates the promising application of the fabricated TDV in the thermal management of advanced packaging systems.

## 1. Introduction

Chip fabrication is continually evolving to accommodate lightweight miniaturization and integration. Based on traditional packaging technologies, new advanced integrations from 2D to 3D structures are also rapidly developing in the semiconductor manufacturing process and microelectronics packaging integration [1,2]. Such integrations have developed more maturely in optoelectronic devices such as CMOS image sensors [3], as well as logic and memory chips [4]. These devices do not usually face severe overheating problems. Nevertheless, the heat dissipation problems of 3D integration must still be considered in other fields, such as high-power electronics, microwave devices, and micro-electromechanical systems [5,6]. A 3D integrated circuit (3D-IC) with through-silicon vias (TSV) is a promising technology to improve device performance while maintaining power consumption because the vertical structure shortens the signal path and reduces the chip interconnect area [7,8]. Thus, it can meet the requirements of high-speed computing, high-density memory, low power consumption, and small package size. However, the higher integration level is accompanied by an increase in power density, thereby resulting in problems such as more severe internal heat storage and prominent local overheating [9,10]. 

Diamond exhibits excellent thermal conductivity (≥2000 W/mK), electrical insulation, high-temperature resistance, and corrosion resistance, thereby efficiently transferring the heat concentrated in devices [11]. Several reported techniques have been developed using diamonds as a hybrid heat spreader or in combination with microfluidics for thermal management [12,13,14]. Among these techniques, bonding closer to the heat junction requires lower thermal interface resistance (TIR) to promote higher 3D heat transfer. 

Advanced 3D integration methods based on Si chips presently include the preparation of TSVs, Cu filling, and hybrid bonding [15,16]. However, few reports have considered the integration of diamonds in 3D-IC. Owing to the difficulties in handling and electroplating thin TSV wafers (<200 μm), dry etching is often used to form blind vias prior to the deposition of insulating, barrier, and seed layers, followed by electroplating or electroless plating to fill the blind holes with metal. Subsequently, wafer thinning and chemical–mechanical polishing processes are used to expose the metal filling to form through vias [17,18,19]. Although dry etching during TSV preparation can be conducted at low temperatures without device damage, the high hardness and stability of diamonds cause various problems, such as low etching efficiency and complex process flow [20]. Unlike Si, uniform thinning is difficult to realize for large-area diamonds. Further, only a few reports have focused on Cu-filled high-aspect-ratio through-diamond vias (TDV) because of the high insulating properties and low metal adhesion of diamonds. 

In this study, we demonstrated an innovative bottom-up Cu plating technology to achieve high-aspect-ratio (10:1) TDV. TDV arrays were fabricated by using ultraviolet (UV) nanosecond laser processing. Subsequently, TDV was bonded to a temporary carrier by the deposition of a metal diffusion layer that serves as the bottom conducting layer. Under different plasma pretreatment conditions, the electrolyte wettability of the diamond and metal surfaces was investigated. Finally, a high-aspect-ratio Cu-filled TDV was achieved through a bottom-up electroplating process. An average single-via resistance of up to ≤50 mΩ was achieved, thereby highlighting the strong potential of TDV for 3D integrated thermal and electrical management applications while exploiting the unique properties of diamond.

## 2. Materials and Methods

Polycrystalline diamond chips were fabricated to a size of 10 × 10 mm^2^ with a thickness of 200 ± 20 μm and polished to achieve an ultra-smooth surface. Subsequently, the through vias structure of the diamond was prepared by using a UV nanosecond laser processing system with a wavelength of 355 nm, a spot size of 12 μm, a pulse width of 12 ns, a repetition rate of 40 kHz, a laser power of 15 W, and a laser duration of approximately 3 s for each hole. The Cu-filling process of the TDV is shown in Figure 1. First, a Cr/Au layer was double-sided deposited by physical vapor deposition. The back side Au layer could serve as the conducting layer in the electroplating process and also as the bonding layer, which was temporarily bonded to the Au-deposited quartz flakes by hot pressing (at 5 MPa and 200 °C for 10 min). The electroplating solution comprised 165 g/L CuSO_4_, 60 g/L H_2_SO_4_, and 0.05 g/L NaCl. In addition, 0.01 g/L bis(3-sulfopropyl) disulfide, 1.5 g/L polyethylene glycol, and 0.02 g/L polyethyleneimine alkyl salt were used as the accelerator, inhibitor, and leveler, respectively. The TDV surface was modified using a plasma treatment at 300 W for 2 min to improve the wettability. The surface was then immersed in deionized water with or without being placed in a vacuum chamber. Subsequently, the chip was statically immersed in the electroplating solution for the bottom-up Cu-filling process. Using a direct-current (DC) source, the current density was adjusted in the range of 0.15–0.6 ASD with an electroplating time of 4–10 h until the outer surface of the vias was completely closed.

TDV morphology and metal layer deposition were characterized using scanning electron microscopy and energy-dispersive X-ray spectroscopy (SEM and EDS, Merlin Compact, ZEISS, Jena, Germany). The improved electrolyte wettability was analyzed by using a contact wetting angle test (OCA20, Dataphysics, Filderstadt, Germany). The Cu-filled TDVs were examined by using computerized tomography (CT, XTH 225, NIKON, Tokyo, Japan). The stress on the diamond surface after Cu deposition was analyzed by using Raman spectroscopy (LabRam HR800, HORIBA Scientific, Palaiseau, France). Moreover, the through-vias tandem line was prepared by magnetron sputtering after lithography before electroplating thickening. Finally, the overall electrical signal was tested using the four-point probe (4PP) method.

## 3. Results and Discussion

### 3.1. Morphology of the TDV

The SEM images of the TDVs at the front and rear surfaces after laser processing are shown in Figure 2A,B. The diamond exhibits a smooth surface, with micro-vias entrance and exit diameters of approximately 27 and 16 μm, respectively, and vias pitch of 55 μm. A slight chipping area with a diameter of approximately 34 μm was noted at the entrance, which can be ascribed to the material ejection during laser removal. Figure 2C shows a schematic of the chip disconnection along the central array of the micropores to obtain a cross-section after the double-sided deposition of the Cr/Au layer. The cross-section of the micro-vias has a flared shape with a depth of approximately 20 μm at the inlet and a rougher surface on the sidewalls [21], which may hinder the practical application of TDVs. A uniform diameter was noted below the entrance to the bottom with a smooth surface and with a minimal taper on the sidewalls, as shown in Figure 2E. These are conducive to the uniform deposition of the seed layer and higher uniformity of the concentrated electric field distribution on the sidewalls during electroplating. In actual applications, the aspect ratio and taper of the micro-vias can be defined according to the upper ($D_{en}$) and lower ($D_{ex}$) pore diameters and diamond thickness (t), which are 10/1 and 1.5°, respectively:(1)$$\mathrm{Aspect}\;\mathrm{ratios}=\frac{D_{en}+D_{ex}}{2t}$$
(2)$$\mathrm{Taper}=\tan^{-1}\left(\frac{D_{en}-D_{ex}}{2t}\right)$$

The EDS elemental distribution at different points in Figure 2C is described in Figure 2D. The flared shape of the taper increased the amount of metal deposited at the entrance. Although the thickness of the metal layer cannot be identified using EDS, the Au content is evenly distributed on the inner wall of the vias, with the lowest Au content in the middle (point 4).

### 3.2. Bottom-Up Cu Electroplating Process 

Figure 3A shows the results of the contact wettability tests for the diamond and Au flat surfaces treated with different plasma conditions activated at 300 W for 2 min and exposed to air for a specific duration. The original flat surfaces of diamond and Au show poor wettability to the electrolyte, and the contact angles are approximately 61° and 82°, respectively. Therefore, the wettability and prewetting treatment before the electroplating process should be improved. Oxygen plasma treatment improved the wettability of the Au surface. In contrast, the Ar plasma treatment showed better results for the diamond surface. This can be attributed to the active groups of the oxygen plasma, which can react to remove residual organic matter on the surface and generate more hydrophilic groups, such as carboxyl and hydroxyl groups [22]. Meanwhile, low ionization energy and heavy atoms increased the energy of Ar during ion bombardment, thereby cleaning and activating the surface. Therefore, Ar/O hybrid plasma treatment was used to improve the wettability of the TDV before electroplating. As shown in Figure 3B,C, the flat diamond surface with weak hydrophilicity (approximately 61°) was transformed into a hydrophilic surface in the TDV (approximately 47°). Hence, the electrolyte drop can diffuse along the top of the vias. The high surface ratio of the TDV with micro-vias structures enhanced the diffusion effect, thereby improving the hydrophilicity [23]. Thus, the integrated effects of the natural wettability and structure asperities of the surface further increased the hydrophilicity of the micro-vias structures by plasma irradiation. 

Figure 4A shows the self-wetting of the electrolyte for TDV with different wetting walls. From the finite element model (FEM) simulation with the vertical insertion at 5 mm/s, the volume fraction of the electrolyte in the vias increased with the decrease in contact angle, and bubbles were observed at the bottom or middle positions. Self-wetting can only be maintained with a small contact wetting angle under fresh plasma treatment; thus, prewetting using ultrasound or vacuum is required [24]. The improper air agitation during the plating process may also cause the formation of bubbles. Figure 4B–D show the 2D-CT scans of the samples after Cu filling at a current density of 0.45 ASD for 6 h after different pretreatments. The Cu filling of the vias exhibited undesirable properties without pretreatment (Figure 4B,C). In particular, obvious closure pinch-off was noted at the top, and the effective filling depth-to-width ratio was approximately 5:1 and 7:1 without pretreatment and only with vacuum-assisted prewetting, respectively. When Ar/O plasma pretreatment was combined with vacuum-assisted prewetting, a Cu-filled TDV was effectively achieved with minimal leakage in the middle, as shown in Figure 4D.

Figure 5A–C, respectively, illustrate the results of the 2D-CT scan after electroplating for 10, 8, and 4 h until the surface was completely closed at current densities of 0.1, 0.3, and 0.6 ASD. The pinch-off phenomenon occurred when the current density was increased to 0.6 ASD because of the excessively high current density with a faster deposition rate. In addition, the local charge concentration and electric field inconsistency between the opening and large bottom due to the high-aspect-ratio structure of the through vias limited the original proportional additive effect [25]. Consequently, Cu ions are deposited on the openings before they are fully diffused to the bottom, forming a seam. As the current density was reduced to 0.3 ASD, the filling condition was significantly improved. The internal 3D-CT diagram is shown in Figure 5D. However, at a lower current density of 0.15 ASD, uneven shrinkage is observed, and seam defects are formed. This can be related to the slow deposition rate and corrosion effect of the solution.

### 3.3. Raman Measurement of Cu-Filled TDV

Figure 6A shows the SEM image of the top surface morphology of the diamond after electroplating, whereby multiple bump electrodes independent of each other were noted. The diameter of the bumps is approximately 35 μm, and the smooth surfaces can be used as interconnection contacts. Raman spectroscopy was used to analyze the stress near the surface of the Cu-filled TDVs. The displacement of the Raman peak with respect to its stress-free frequency provides information on the stresses present in the sample, as follows [26]:(3)$$\nu_{r}-\nu_{s}=\alpha \sigma$$
where $\nu_{r}$ is the position of the stress-free Raman line obtained by a multi-point measurement of the TDV surface before Cu filling (1332.5 ± 0.1 cm^−1^), $\nu_{s}$ is the measured frequency of the Raman line, σ is the stress present (GPa), and α is the stress-induced frequency shift factor with an average value of 2.88 ± 0.17 cm^−1^/GPa. The compressive stress shifted the peak to higher frequencies, whereas the tensile stress shifted it to lower frequencies.

As the measurement position approaches the Cu-filled TDV, the Raman peak position shifts toward the compression side. The compact filling and coefficient of thermal expansion difference between bulk Cu and diamond increased the compressive stresses at the vias edges. Moreover, the differences in the internal grain size and radial distribution owing to the use of polycrystalline diamond led to uneven stress distribution. Thus, reliability issues due to the large material property differences between diamond and Cu should be considered.

### 3.4. Resistance Measurement of the Cu-Filled TDV

The resistance of through vias is a key parameter for evaluating their electrical performance. By preparing a serpentine through vias to interconnect the structure on the Cu redistribution line, the Kelvin 4PP method was used to measure the resistance of chain 1 formed by 100 through vias in series [27], and chain 2 of equal length to chain 1. The sample was achieved by PVD alignments after lithography and before electroplating thickening, as shown in Figure 7A. The test results are shown in Figure 7B. The I–V curves show that the total resistances of chain 1 and the interconnection line chain 2 are 9.63 and 4.88 Ω, respectively. The average DC resistance of the Cu-filled single via was calculated to be approximately 47.5 mΩ. This low interconnection resistance promotes the development of vertical structures for various electronic components.

### 3.5. Thermal Analysis Simulation

The relevant thermal properties of the system were evaluated by solving the steady-state heat conduction equation using FEM and implemented by the COMSOL Multiphysics 5.6 package. Figure 8A shows the Cu-filled chip model used to evaluate the equivalent thermal conductivity. The thermal conductivity is anisotropic owing to the Cu filling, i.e., the equivalent thermal conductivity in the x-/y-plane direction ($k_{eq,x}=k_{eq,y}$) is not equal to that in the vertical direction ($k_{eq,z}$). These can be extracted by the following equation [28]:(4)$$k_{eq,z}=Q\frac{\Delta z}{\left|\Delta T\right|}$$
(5)$$k_{eq,x}=Q\frac{\Delta x}{\left|\Delta T\right|}$$
(6)$$k_{eq,x}=k_{eq,y}$$
where Q is the input heat, Δx and Δz are the distances in the horizontal and vertical heat conduction directions, respectively, and ΔT is the temperature difference at both ends.

The calculated equivalent thermal conductivity of TDV and TSV for different pitches are shown in Figure 8B. With the pitch decreased, the Cu-filled volume fraction increased without changing the size of the vias. Diamond (1500 W/mK) has an ultra-high thermal conductivity, whereas Si (150 W/mK) has a lower thermal conductivity than Cu (400 W/mK); these materials have opposite Cu-filling trends. It is worth noting that the Cu filling has a larger effect on the in-plane thermal conductivity than that on the out-plane thermal conductivity due to the discontinuous phase. 

Furthermore, simulations were performed to evaluate the TDV as a heat spreader layer that should be integrated with TSV under different design parameters, as shown in Figure 8C. The simulated cooling package (2.5 × 2.5 mm^2^) includes TSV chip layer 1 (20–200 μm thickness) and TDV chip layer 2 (200 μm thickness). The area containing through vias is limited to 2 × 2 mm^2^. All vias are filled with bulk Cu with a diameter of 20 μm. The TIR between chip layers 1 and 2 is 0.5 mm^2^ K/W, based on studies on relevant 3D bonding interfaces [16]. The Newtonian cooling process was enhanced by the TDV chip backside surface with an effective heat transfer coefficient of 15 kW/m^2^K at an ambient temperature of 20 °C. The hot spot was simulated by attaching a heat source to the upper surface of chip 1 with a heat flux and size of 1 kW/cm^2^ and 1 × 1 mm^2^, respectively. The initial temperature of all parts matches that of the environment. To conservatively predict the maximum temperature and thermal resistance, adiabatic conditions were assumed on all remaining exposed surfaces of chip 1 and 2.

The variations in the maximum surface heat source temperature with the pitch and Si chip thickness were investigated, as shown in Figure 8D–F. The hot spot temperature significantly increased with increasing pitch, i.e., when a 200-μm-thick TSV replaced the bottom TDV (green line) to bond with the 50-μm-thick upper TSV chip 1. The highest temperature was calculated to be 170.0 °C at a pitch of 150 μm, indicating that the low thermal conductivity of Si severely limited the performance of the device in 3D stacks. Figure 8D,E show the thermal distribution of the package structure with pitches of 50 and 100 μm, respectively. Owing to the low thermal conductivity limitation of Si, the Cu filling of the narrow pitch has a larger thermal benefit, and the temperature of the hot spot greatly decreased to approximately 5 °C. Comparing the red and green lines in Figure 8F, when a 200-μm-thick TDV was integrated with the 50-μm-thick upper TSV, the hot spot temperature was effectively suppressed owing to the high thermal conductivity of the diamond (approximately 20 °C). However, when the thickness of the upper TSV chip was decreased from 200 μm to 20 μm, the hot spot temperature decreased with the increase in the through via pitches, which is ascribed to the extremely thin Si chip layer and thermal resistance of the bonding interface. The homogeneous temperature effect of the diamond layer started to dominate. Combined with Figure 8B, the increase in equivalent thermal conductivity increased the heat dissipation gain, which is less affected by the distribution condition of the through vias. Finally, our simulation is only a simple demonstration of the back-to-back stacking structure of two-layer chips. For multi-layer chip packaging, more factors should be considered, such as the thickness of each layer, the horizontal layout of hot spots, and size.

## 4. Conclusions

This study demonstrated the fabrication of an innovative Cu-filled TDV structure using UV nanosecond laser drilling. The longitudinal average diameter of diamond vias is approximately 20 μm with an aspect ratio of 10:1. Through a PVD method, a uniform Cr/Au conductive layer was double-sided deposited on the sidewalls, which was then bonded to the temporary substrate by the retained rear surface metal layer. Before the Cu filling, the electrolyte wettability of the diamond and metal surfaces was simultaneously improved by the synergistic Ar/O plasma activation treatment. The bottom-up wet electroplating technology was adopted to realize the Cu-filled TDV with a high-aspect ratio (10:1) at a current density of 0.3 ASD. The results of non-destructive CT tests and DC resistance tests show a good Cu-filling effect. The low average single-via DC resistance of 47.5 mΩ suggests the applicability of this structure for the vertical interconnection of various electronic components. In addition, the relevant thermal properties of simple back-to-back stacking two-layer chips were evaluated by finite element analysis. Especially for thinner chips, the integration of TDV with low thermal resistance will be more beneficial. Therefore, this study provides a promising approach for the development of diamond substrates or advanced 3D integrated packaging in thermal management, especially for high-power devices.

## Figures and Tables

**Figure 1 micromachines-14-00290-f001:**
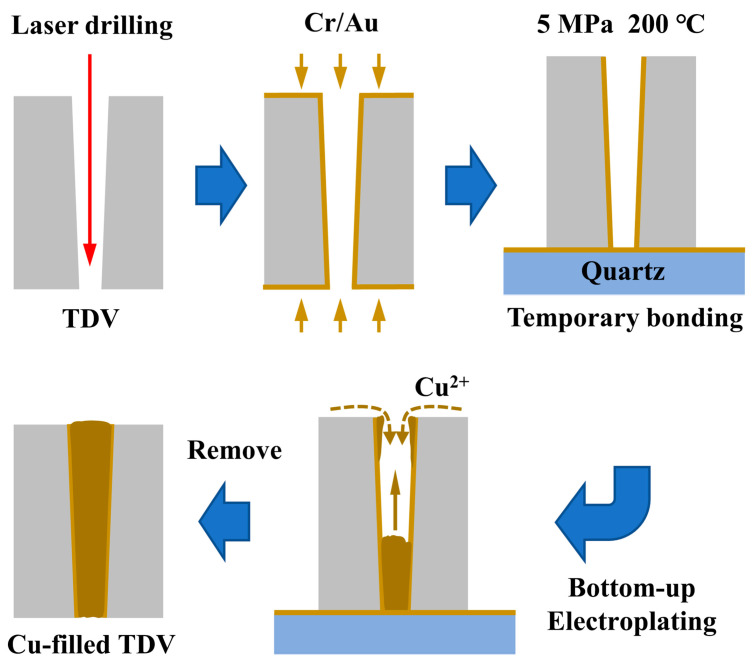
Schematic of the fabrication process of the Cu-filled TDV.

**Figure 2 micromachines-14-00290-f002:**
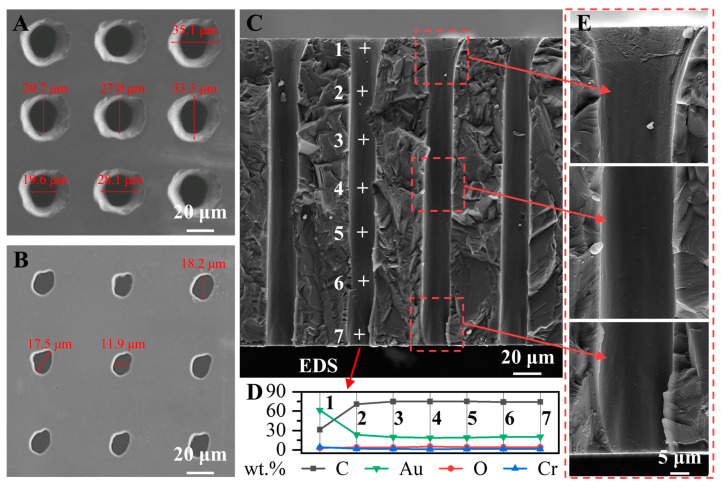
(**A**) Front and (**B**) rear surface SEM images of the TDV. (**C**) Cross-sectional SEM image of TDV. (**D**) EDS points of the vias in (**C**). (**E**) Partial enlargement of (**C**).

**Figure 3 micromachines-14-00290-f003:**
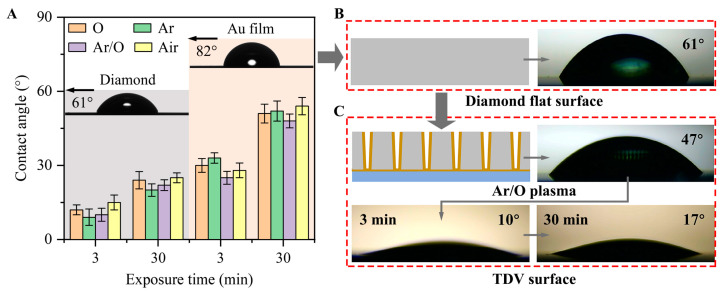
(**A**) Contact angles of the electrolyte on the diamond and Au flat surfaces under short-term exposure to air after plasma irradiation. Droplet shape on different diamond patterns, (**B**) flat diamond surface without plasma treatment, and (**C**) TDV surface without plasma treatment and short-term exposure to air after Ar/O plasma irradiation after Cr/Au deposition and temporary bonding.

**Figure 4 micromachines-14-00290-f004:**
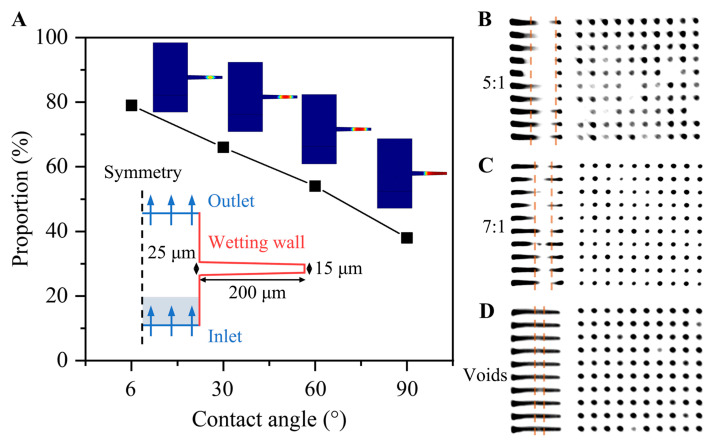
(**A**) Relationship between the contact angle of the wetting wall and self-wetting volume proportion of the electrolyte in TDV. (**B**–**D**) 2D-CT scans of the Cu-filled TDV at 0.45 ASD without any pretreatment, with vacuum-assisted prewetting treatment, and with Ar/O plasma-combined vacuum-assisted prewetting, respectively.

**Figure 5 micromachines-14-00290-f005:**
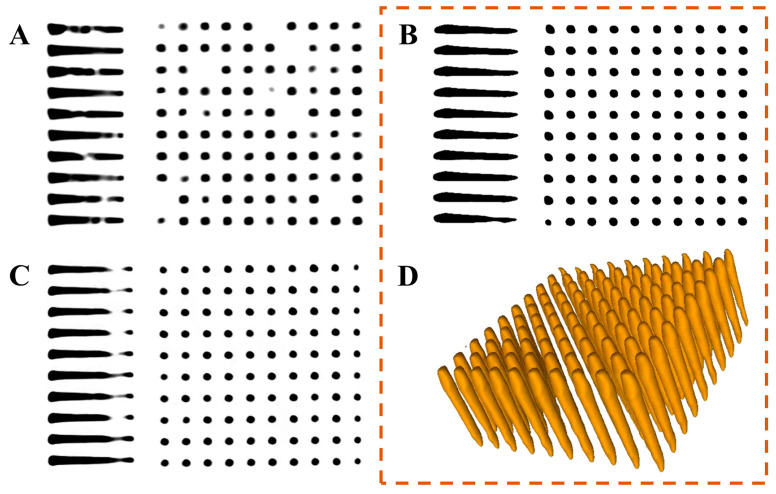
(**A**–**C**) 2D-CT scans of the TDV after complete Cu filling at current densities of 0.15, 0.3, and 0.6 ASD, respectively. (**D**) 3D-CT scan structure of (**B**).

**Figure 6 micromachines-14-00290-f006:**
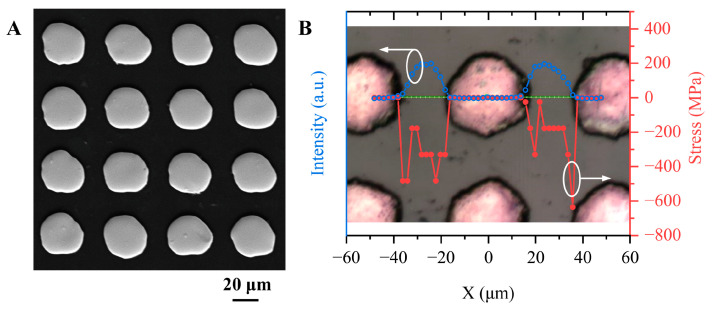
(**A**) SEM morphology of the TDV surface after complete electroplating. (**B**) Stress distribution of the diamond around the Cu-filled TDV front surfaces according to the Raman measurement.

**Figure 7 micromachines-14-00290-f007:**
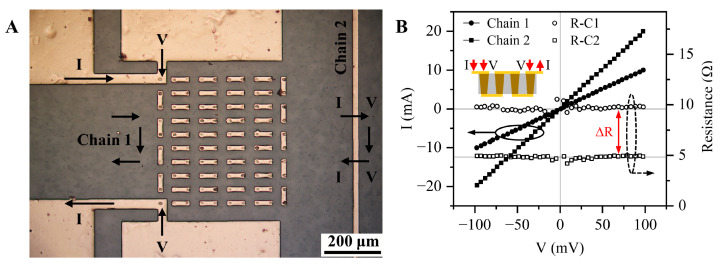
(**A**) Sample with a serpentine series circuit on the surface of both sides of the TDV containing 100 through vias. (**B**) Test results of the I–V curves obtained by the 4PP method and the corresponding DC resistance.

**Figure 8 micromachines-14-00290-f008:**
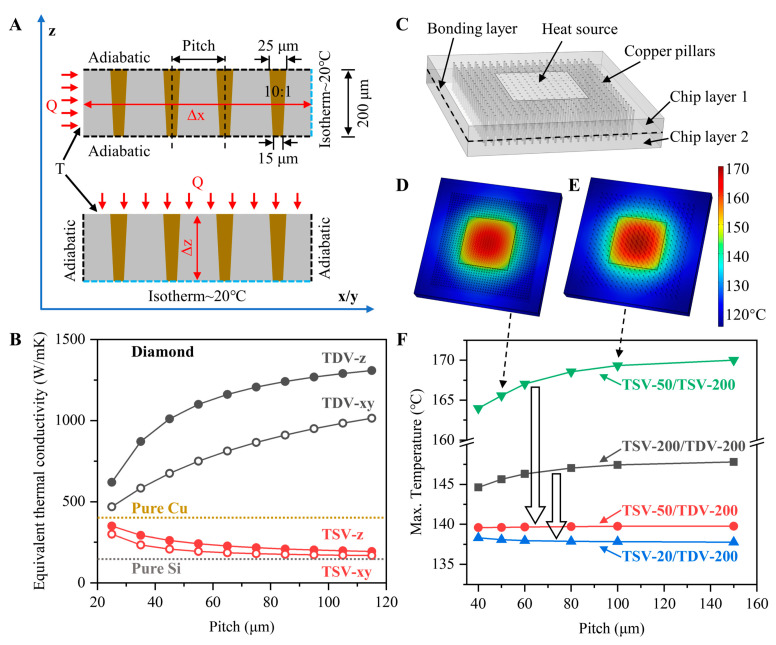
(**A**) Equivalent thermal conductivity model of the Cu-filled chip. (**B**) Relationship between the Cu-filled TSV/TDV anisotropic thermal conductivity and pitches. (**C**) Schematic of the TSV/TDV chip cooling package. Modeled temperature distribution for the two Cu-filled TSV chips with the pitch of (**D**) 50 μm and (**E**) 100 μm, respectively. (**F**) Relationship between the maximum temperature and different pitches with different chip layer thicknesses.

## Data Availability

All data needed to evaluate the conclusions in the paper are present in the paper. Additional data related to this paper may be requested from the authors.

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
