# Peer review of "Bottom-Up Cu Filling of High-Aspect-Ratio through-Diamond vias for 3D Integration in Thermal Management"

_micromachines, 2023, doi:10.3390/mi14020290_

Round 1
Reviewer 1 Report
Please read carefully the conclusions. It is important to extend this.
Please read carefully the results. Please explain.
The chapter 4 is too short. Please extend.
Thank you.
Author Response
Response: Thank you for your pertinent review of our paper. According to your suggestion, we have added more details to the Results and Discussion section and enriched the Conclusions section. In addition, the paper has been thoroughly edited by a professional English editing service to polish the language and ensure that the revised manuscript is free of grammatical or typographical errors.
Line 297–313:
- Conclusions
This study demonstrated the fabrication of an innovative Cu-filled TDV structure by UV nanosecond laser drilling. The longitudinal average diameter of diamond vias is approximately 20 μm with an aspect ratio of 10:1. Through a PVD method, a uniform Cr/Au conductive layer was double-sided deposited on the sidewalls, which was then bonded to the temporary substrate by the retained rear surface metal layer. Before the Cu filling, the electrolyte wettability of the diamond and metal surfaces was simultaneously improved by the synergistic Ar/O plasma activation treatment. The bottom-up wet electroplating technology was adopted to realize the Cu-filled TDV with a high-aspect-ratio (10:1) at a current density of 0.3 ASD. The results of non-destructive CT tests and DC resistance tests show a well Cu filling effect. The low average single-via DC resistance of 47.5 mΩ suggests the applicability of this structure for the vertical interconnection of various electronic components. In addition, the relevant thermal properties of simple back-to-back stacking two-layer chips were evaluated by finite element analysis. Especially for thinner chips, the integration of TDV with low thermal resistance will be more beneficial. Therefore, this study provides a promising approach for the development of diamond substrates or advanced 3D integrated packaging in thermal management, especially for high-power devices.
Reviewer 2 Report
Review of Bottom‐up Cu filling of high‐aspect‐ratio through‐diamond vias for 3D integration in thermal management
In this manuscript, the authors reported a bottom‐up Cu electroplating technique with a high‐aspect‐ratio through‐diamond vias (TDV). Overall, the technique design was well explained. My concerns/comments are as follows.
It seemed that the main motivation of the TDV structure was for heat dissipation and thermal management. While the electroplating techniques were presented reasonably well, the thermal analysis/simulation was not and in fact it was too superficial. The finite element method used to produce the equivalent thermal conductivity was not even briefly introduced. What was the model set up, boundary conditions and initial condition etc.? Was the model validated in any way? Was it possible to perform experiments to investigate thermal performance? After all, since the authors claimed this was an innovative technique, its performance and benefits should be better evaluated.
Author Response
Response: Thank you for pointing out these deficiencies. We improved the establishment of the model in the revised manuscript.
Line 236–239: The relevant thermal properties of the system were evaluated by solving the steady-state heat conduction equation by the finite element model using the COMSOL Multiphysics 5.6 package.
Line 262-273: The simulated cooling package (2.5 × 2.5 mm2) includes TSV chip layer 1 (20–200 μm thickness) and TDV chip layer 2 (200 μm thickness). The area containing through vias is limited to 2 × 2 mm2. All vias are filled with bulk Cu with a diameter of 20 μm. The TIR between chip layers 1 and 2 is 0.5 mm2K/W based on studies on relevant 3D bonding interfaces. Moreover, the Newtonian cooling process was enhanced by the TDV chip backside surface with an effective heat transfer coefficient (HTC) of 15 kW/m2K at an ambient temperature of 20 °C. The hot spot was simulated by attaching a heat source to the upper surface of chip 1 with a heat flux and size of 1 kW/cm2 and 1 × 1 mm2, respectively. The initial temperature of all parts matches that of the environment. To conservatively predict the maximum temperature and thermal resistance, adiabatic conditions were assumed on all remaining exposed surfaces of chip 1 and 2.
Line 292-295: It should be emphasized that our simulation is only a simple demonstration of the back-to-back stacking structure of two-layer chips. For multi-layer chips packaging, more factors should be considered, such as the thickness of each layer, the horizontal layout of hot spots, and size.
Considering the difficulty of the subsequent chip integration process, specific experiments on the actual heat dissipation benefits are difficult at this stage, but we consider this to be an important objective for our future work.
Reviewer 3 Report
In this work, the authors demonstrated a Cu-filled high-aspect-ratio through-diamond vias (TDV) structure for heat dissipation. The average single-via 20 resistance was measured to be ≤ 50 mΩ. This TDV structure and fabrication process can be of interest to the researchers in the field of thermal management in chip packaging. However, the manuscript includes some parts that should be revised and clarified before publication. The authors have to address the following questions and comments and modify their manuscript accordingly.
1) In introduction, the authors write: “…Based on traditional packaging technologies, new advanced integrations from 2D to 3D structures are also rapidly developing, particularly in high-power electronics, microwave devices, and micro-electromechanical systems [1,2]. A 3D integrated circuit (3D‐IC) with through‐silicon vias (TSV) is a promising technology for improving performance without increasing power consumption because the vertical structure shortens the signal path and reduces the chip interconnect area [3,4]…”. The introduction has much room to be improved. Advanced integrations from 2D to 3D structures have also been applied in optoelectronic devices. It would be great if the authors include these developments and achievements in the introduction, so to give the readers a much broader view. Several recent references, such as such as Optics Express 27(12), A669 (2019); Laser & Photonics Reviews 2022, 2200455 (https://doi.org/10.1002/lpor.202200455); Optics Express 25(22), 26615 (2017), etc., etc. should be added, so that the readers can be clear about the state-of-the-art of this topic.
2) The through-vias structure of the diamond was fabricated by a UV nanosecond laser. As the parameters of the laser source influence the processing effect of through-vias, the authors should provide the wavelength, frequency, spot size et al. of the UV laser source.
3) As shown in Figure 2c, the micro-vias exhibited rough sidewall and the surface topography is inhomogeneous. Have the authors considered how to improve the fabrication process?
4) The finite element model simulation software used in this work should be mentioned and please state the assumption of simulation to cater the non-idea setup.
5) In the 3.2. Bottom‐up Cu electroplating process section, I suggest that labelling the effective filling depth-to-width ratio in Figure 4 will improve the readability.
6) The authors investigated the effect of Cu filling in micro-vias by computerized tomography. It is suggested to provide SEM images of TDV after Cu electroplating process, which can be more straightforward and convincing.
7) “Raman Measurement of Cu‐filled TVD” should be corrected to be “Raman Measurement of Cu‐filled TDV”; “Resistance Measurement of the Cu‐filled TVD” should be corrected to be “Resistance Measurement of the Cu‐filled TDV”.
Author Response
Response 1: We agree that the Introduction needed major improvements. We have revised the manuscript according to your suggestions. 3D integration technology plays an important role in semiconductor manufacturing and back-end packaging integration. It has developed more maturely in optoelectronic devices such as CMOS image sensors, as well as logic and memory chips. Such devices do not face severe overheating problems. Nevertheless, the heat dissipation problems of 3D integration must still be considered in more fields, such as in high-power electronics, microwave devices, and micro-electromechanical systems.
Line 29–37: Based on traditional packaging technologies, new advanced integrations from 2D to 3D structures are also rapidly developing in the semiconductor manufacturing process and microelectronics packaging integration [1,2]. Such integrations have developed more maturely in optoelectronic devices such as CMOS image sensors [3], as well as logic and memory chips [4]. These devices do not usually face severe overheating problems. Nevertheless, the heat dissipation problems of 3D integration must still be considered in other fields, such as high-power electronics, microwave devices, and micro-electromechanical systems [5,6].
- Zhang, S. Review of Modern Field Effect Transistor Technologies for Scaling. J. Phys. Conf. Ser. 2020, 1617, doi:10.1088/1742-6596/1617/1/012054.
- Ko, C.T.; Chen, K.N. Wafer-level bonding/stacking technology for 3D integration. Microelectron. Reliab. 2010, 50, 481–488, doi:10.1016/j.microrel.2009.09.015.
- Zhou, S.; Liu, X.; Yan, H.; Chen, Z.; Liu, Y.; Liu, S. Highly Efficient GaN-Based High-Power Flip- Chip Light-Emitting Diodes. Opt. Express 2019, 27, 669–692, doi: 10.1364/OE.27.00A669.
- Li, C.; Wang, X.; Song, S.; Liu, S. 21-layer 3D chip stacking based on Cu-Sn bump bonding. IEEE Trans. Componen. Packag. Manuf. Technol. 2015, 5, 627–635, doi:10.1109/TCPMT.2015.2418274.
- Liu, B.; Bi, T.; Fu, Y.; Kudara, K.; Imanishi, S.; Liu, K.; Dai, B.; Zhu, J.; Kawarada, H. MOSFETs on (110) C–H Diamond: ALD Al2O3/Diamond Interface Analysis and High Performance Normally-OFF Operation Realization. IEEE Trans. Electron Devices. 2022, 69, 949-955, doi: 10.1109/TED.2022.3147152.
- Wang, Z. Microsystems Using Three-Dimensional Integration and TSV Technologies: Fundamentals and Applications. Microelectron. Eng. 2019, 210, 35–64, doi:10.1016/j.mee.2019.03.009.
Response 2: In this study, various laser source parameters and material properties significantly impact the processing of through vias. We have added more details on the experimental parameters to discuss these effects further.
Line 79–82: Subsequently, the through-vias structure of the diamond was prepared by a UV nanosecond laser processing system, with a wavelength of 355 nm, spot size of 12 μm, pulse width of 12 ns, repetition rate of 40 kHz, laser power of 15 W, and laser duration of approximately 3 s for each hole.
Response 3: Due to the difficulty of diamond processing, we tried various modulated laser sources and different equipment. Finally, we determined that an ultraviolet nanosecond laser is effective in processing the diamonds in this study. We obtained high-quality diamond micro vias with a high-aspect-ratio (10:1). To further improve the high-quality diamond micro vias, a sacrificial layer may be added to protect the surface and reduce drilling efficiency. A femtosecond laser may also be used.
Response 4: We added more details about the finite element modelling software to better establish the model in this study.
Line 236–239: The relevant thermal properties of the system were evaluated by solving the steady-state heat conduction equation using FEM and implemented by the COMSOL Multiphysics 5.6 package.
Line 262–273: The simulated cooling package (2.5 × 2.5 mm2) includes TSV chip layer 1 (20–200 μm thickness) and TDV chip layer 2 (200 μm thickness). The area containing through vias is limited to 2 × 2 mm2. All vias are filled with bulk Cu with a diameter of 20 μm. The TIR between chip layers 1 and 2 is 0.5 mm2K/W, based on studies on relevant 3D bonding interfaces [16]. The Newtonian cooling process was enhanced by the TDV chip backside surface with an effective heat transfer coefficient (HTC) of 15 kW/m2K at an ambient temperature of 20 °C. The hot spot was simulated by attaching a heat source to the upper surface of chip 1 with a heat flux and size of 1 kW/cm2 and 1 × 1 mm2, respectively. The initial temperature of all parts matches that of the environment. To conservatively predict the maximum temperature and thermal resistance, adiabatic conditions were assumed on all remaining exposed surfaces of chip 1 and 2.
Line 292–295: It should be emphasized that our simulation is only a simple demonstration of the back-to-back stacking structure of two-layer chips. For multi-layer chips packaging, more factors should be considered, such as the thickness of each layer, the horizontal layout of hot spots, and size.
Response 5: Thank you for this suggestion. We have added such details and definitions to enhance the readability of the revised manuscript. The sizes of the model in Figure 4 are also labelled accordingly.
Response 6: We agree that the SEM images would be an interesting addition. We have tried to obtain the cross-sectional SEM images of Cu-filled TDV by various methods. However, the ablation and material splash in the laser cutting process seriously damaged the structure. In addition, owing to the significant difference in hardness between diamond and Cu, we could not obtain a complete cross-section using ion polishing and FIB. Therefore, we used non-destructive testing (CT), which is also widely used in the industry, to evaluate the Cu filling. However, we understand that some details can be left undetected in CT measurements.
Response 7: The spelling errors have been corrected. We have also carefully checked the revised manuscript for such errors.
Round 2
Reviewer 2 Report
The revised version is acceptable.